# Anthropogenic Influence on Moth Populations: A Comparative Study in Southern Sweden

**DOI:** 10.3390/insects14080702

**Published:** 2023-08-11

**Authors:** Markus Franzén, Anders Forsman, Bafraw Karimi

**Affiliations:** Department of Biology and Environmental Science, Linnaeus University, 391 82 Kalmar, Sweden; anders.forsman@lnu.se (A.F.); bk222dz@student.lnu.se (B.K.)

**Keywords:** abundance, anthropogenic effects, community composition, environmental changes, flight period changes, insect conservation, moth populations, range shifts, species richness, southern Sweden

## Abstract

**Simple Summary:**

This research investigates moth biodiversity in two southern Swedish provinces, Västergötland and Småland, spanning from 1974 to 2019. The moth diversity over these years was evaluated using data collected from literary sources. To augment this dataset, a light trap was installed in each province in 2020. The data demonstrate enhanced diversity in Kalmar, Småland, and a more rapid colonisation rate throughout the study period in Småland compared to Västergötland. Noteworthily, our traps in Västergötland and Småland captured 44% and 28% of the known moth species in these provinces, respectively. We reveal significant associations between the probability of species presence in the traps and specific traits when contrasted with a provincial species pool. Traits disproportionately represented in the traps encompass species with considerable variation in colour patterns, generalist habitat and host plant preferences, extended flight periods, and species that primarily overwinter as eggs. This research underscores the influences of climate change and human activities on the shaping of moth biodiversity.

**Abstract:**

As moths are vital components of ecosystems and serve as important bioindicators, understanding the dynamics of their communities and the factors influencing these dynamics, such as anthropogenic impacts, is crucial to understand the ecological processes. Our study focuses on two provinces in southern Sweden, Västergötland and Småland, where we used province records from 1974 to 2019 in combination with light traps (in 2020) to record the presence and abundance of moth species, subsequently assessing species traits to determine potential associations with their presence in anthropogenically modified landscapes. This study design provides a unique opportunity to assess temporal changes in moth communities and their responses to shifts in environmental conditions, including anthropogenic impacts. Across the Västergötland and Småland provinces in Sweden, we recorded 776 moth taxa belonging to fourteen different taxonomic families of mainly Macroheterocera. We captured 44% and 28% of the total moth species known from these provinces in our traps in Borås (Västergötland) and Kalmar (Småland), respectively. In 2020, the species richness and abundance were higher in Borås than in Kalmar, while the Shannon and Simpson diversity indices revealed a higher species diversity in Kalmar. Between 1974 and 2019, the colonisation rates of the provinces increased faster in Småland. Ninety-three species were found to have colonised these provinces since 1974, showing that species richness increased over the study period. We reveal significant associations between the probability of a species being present in the traps and distinct traits compared to a provincial species pool. Traits over-represented in the traps included species with a high variation in colour patterns, generalist habitat preferences, extended flight periods, lower host plant specificity, and overwintering primarily as eggs. Our findings underscore the ongoing ecological filtering that favours certain species-specific traits. This study sheds light on the roles of climate change and anthropogenic impacts in shaping moth biodiversity, offers key insights into the ecological processes involved, and can guide future conservation efforts.

## 1. Introduction

Human influence and climate change are driving substantial changes in the species compositions of habitats and ecosystems [1,2,3]. Understanding the biodiversity of complex and rich insect communities is an essential cornerstone of ecological research [4,5]. The order Lepidoptera, commonly known as moths and butterflies, offers key insights into ecosystem health and resilience [6]. Studying how changes in the distribution, diversity, and abundance of moths are influenced by environmental changes and disturbances and how the responses are modified by species traits can yield significant insights into how biodiversity will respond to human activities in the future [7,8]. Current environmental changes impose intense pressure on species, especially those with narrow and specific ecological requirements, leading to decreased population sizes, shrinking distribution ranges, and increased extinction rates [9,10,11]. In contrast, generalist species, characterised by broader ecological tolerance, demonstrate remarkable resilience by being able to exploit available resources and adapt their phenology and geographical range to suit the present circumstances [1,12,13,14].

Many moth species are expanding their range limits northwards and exhibit changes in their population numbers and adult activity periods [2,15,16,17]. It is hypothesised that recent climate warming has influenced these shifts, particularly affecting the length of the adult flight period, the peak day of activity, and the range size [18]. However, the impact of these ecological alterations varies among species, underlining the necessity for in-depth analysis. For instance, understanding what determines colonisation rates and the abundance of recently colonising species at a local scale is crucial. It can be hypothesised that newly colonised species might be rare due to limited host plant availability [19] or unsuitable climatic conditions at the range margin [20]. Conversely, these range-expanding species can potentially capitalise on newly available resources and become abundant [21,22]. Another compelling aspect to consider is whether the compositions of local species communities in anthropogenically influenced sites are predictable based on their trait distributions, compared to the distribution of traits of potential colonisers in the surrounding species pool [9,13]. In contemporary landscapes that are heavily influenced by human activity, generalist species are likely to predominate [23].

In this study, we combine two unique moth datasets to investigate whether and how changes in the species richness, composition, diversity, and abundance of moths are influenced by environmental conditions and species traits. The first dataset encompasses local data from two light traps situated at two sites in southern Sweden, collected throughout 2020. The two study sites, Borås, dominated by spruce plantations, and Kalmar, exhibiting a more urban environment, hold distinct landscapes that could affect moth communities differently based on their ecological needs. However, given that all land within a 1 km radius of the traps is anthropogenically influenced and unprotected, it is also plausible that moth communities might exhibit similar responses across these landscapes due to their shared human-induced modifications. The second dataset spans larger-scale provincial data, covering all species logged in the two provinces where the traps were located, along with the inaugural year that each species was observed in the respective province, tracing back to 1974. The two provinces differ in several important respects, viz. climate, human density, and land use, which are likely to affect the ecological success of moths depending on their ecological requirements. To evaluate the role of ecological filtering, we compiled, for each species, information on five different traits (length of flight period, habitat preference, overwintering stage, host plant specificity, and adult colour pattern variation; for details see Methods, Section 2.2). These traits capture different dimensions of the niche breadth and ecological generalisation in moths [12] and have previously been shown to be associated with high establishment success and the ability to cope with novel, variable, and changing environmental conditions [9,17,24,25]. The study design thus enables us to address the following four questions:Do species richness, abundance, and diversity measurements differ between the provinces and local trapping sites?Does the community composition vary between the two trapping sites, and do the provincial colonisation rates differ between the two provinces?Is there a positive or negative correlation between the years since provincial colonisation and the abundance of moth species found in the local traps? Such a correlation might be expected if species rapidly become abundant after colonisation and subsequently decrease or if species gradually increase in number over time. Are recently arriving species to the provinces rare, expansive, or potentially non-native invaders that have not previously reported in Sweden?Are certain species traits more likely to be associated with moths in anthropogenic landscapes (species present in the traps) compared to those in the surrounding species pool (species known to occur in the provinces).

To address these questions, we conducted an exhaustive analysis involving 776 species registered across the two provinces. This investigation was augmented by data from two light traps strategically located in both provinces. These traps, operational throughout 2020, successfully captured a remarkable 27,797 individuals, of 371 moth taxa.

## 2. Material and Methods

### 2.1. Study Sites

The study was undertaken in the southern regions of Sweden, specifically in the provinces of Västergötland and Småland. Västergötland spans an area of 16,676 km^2^ and is home to 1.4 million inhabitants. Its geographical features include a western coastal environment characterised by a chain of islands and skerries in the Skagerrak Sea. Progressing inland, the landscape subtly transforms into a gently undulating countryside punctuated by numerous lakes and a blend of deciduous and coniferous forests. In contrast, Småland, covering 30,689 km^2^ and hosting a population of 750,000, is primarily dominated by a forested high plain. The soil composition, a mix of sand and boulders, renders the land largely unsuitable for agricultural practices, with the exceptions being the Kalmar plains and select coastal regions. The province is also interspersed with lakes and bogs. The coastline presents a varied landscape, featuring bays in the north, an offshore island archipelago, and cultivated flatlands in the south.

Both provinces are characterised by a seasonal climate. Summer (June, July, and August) typically averages temperatures of around 20 °C, while winter (December, January, and February) averages 0 °C. On the coldest days, temperatures can plunge to as low as −10 °C. Precipitation, moderately distributed throughout the year, is the highest during October and November. Both regions receive annual precipitation levels ranging from 600 to 800 mm, a significant portion of which falls as snow in the winter [26]. Winds, primarily from the west, sporadically influence the area, bringing mild, humid air masses that affect the local temperature and precipitation patterns [27]. 

One of the authors (BK) deployed one automatic Ryrholm light trap at one site near Borås (57°36′13.61″ N, 12°46′47.11″ E; in the province of Västergötland) and one trap in Kalmar (56°41′12.43″ N, 16°19′53.20″ E; the province of Småland). The Euclidian distance between the traps was 235 km. The Borås trap was powered by a 250 W mercury lamp (Sylvania HSL-GW, Osram Sylvania, Wilmington, MA, USA), and the Kalmar trap was powered by two 40 W ultraviolet fluorescent tubes (Wemlite LT40WX, Wemlite, Andover, United Kingdom) [28]. Previous studies have demonstrated that these apparatuses are of similar efficiency and are highly effective for monitoring moths [29,30]. The lamps were connected to twilight sensors for autonomous operation and programmed to switch on at dawn and off at dusk. The traps were emptied every ten to twelve days. The operation period for the traps extended from 12 March 2020 to 29 November 2020 at the Borås site and from 13 March 2020 to 23 November 2020 at the Kalmar site, yielding 252 and 255 sampling nights per site, respectively. Macro-moths were identified at the species level by BF using an identification guide [31], with the identification process validated by MF. The following 14 moth families were studied: Brahmaeidae, Cossidae, Drepanidae, Endromidae, Erebidae, Geometridae, Hepialidae, Lasiocampidae, Limacodidae, Noctuidae, Nolidae, Notodontidae, Saturniidae, and Sphingidae (Appendix A). Some individuals could not be definitively identified at the species level. Consequently, they were classified at the species complex level, leading to a pooling of 18 species into nine taxa. A total of 188 individuals were identified as belonging to these combined taxa (Appendix A). The species taxonomy follows Aarvik, et al. [32] and red-listed species follow Eide, et al. [33]. 

These traps were strategically placed at forest edges within anthropogenically influenced landscapes that were void of original natural habitats within a 3.1 km^2^ area (1 km radius). Despite differences in land cover, these sites are representative of the characteristic human-modified landscapes in southern Sweden. The landscape surrounding the Borås trap primarily comprises spruce plantations (79%), open grasslands (10%), built-up areas or gardens (10%), and deciduous forests or parks (1%). Conversely, the landscape surrounding the Kalmar trap is dominated by gardens and built-up areas (80%), along with deciduous forests (10%), coniferous forests (1%), open grasslands (4%), and parks (5%). The data above were extracted from the Swedish land cover data [34].

### 2.2. Characterisation of Species Traits

In this study, we collected data on five key traits for the 776 species recorded in the two provinces, which included the 371 species identified in the traps. These traits are the length of flight period, habitat preference, overwintering stage, host plant specificity, and adult colour pattern variation, and they capture different aspects of the niche breadth and ecological generalisation in moths [12]. The data for these traits were derived from various sources [35,36,37,38,39,40]. The length of flight period, treated as a continuous variable, encompasses the entire period of adult activity. Habitat preference was categorised into three distinct types: ‘forest’, ‘open’ (encompassing shrublands, wetlands, grasslands, and other open spaces), and ‘generalist’, with the latter term used for species found across various habitats. The overwintering stage was classified into four categories: ‘egg’, ‘larva’, ‘pupa’, and ‘imago’. Host plant specificity was treated as a continuous variable based on the number of plant species, genera, or families consumed during the larval stage. Here, specialists scored as 1, predominantly fed on a single plant species. Oligophagous species, scored as 2, consumed fewer than six species or were limited to a specific plant genus/family, and generalist species, scored as 3, fed on six or more plant species or genera. Lastly, we used information on the colour pattern variation. This was because theory and empirical evidence largely concur that populations and species with polymorphic or variable colour patterns have broader niches and are better able to cope with challenges associated with environmental heterogeneity and change [9,12,17,24,25,41,42]. There is also evidence that colour pattern variation can reduce susceptibility to predation [43]. For the purpose of this study, colour pattern variation was treated as a continuous variable, evaluated based on the variation in the colour and pattern of adult moth wings within a species [12,24,44]. ‘Highly variable’ species, scored as 2, exhibited significant differences in the size, shape, and colour of the pattern elements or the presence/absence of these elements. ‘Variable’ species, scored as 1, displayed variations in the pattern element size, shape, or colour. ‘Non-variable’ species, scored as 0, showed no discernible variation in the wing colour or pattern.

### 2.3. Data at the Provincial Level

We downloaded a revised version of the Catalogue of Swedish Lepidoptera [45] on 1 April 2022, which served as the primary source of provincial data for the provinces where the two study sites were located. This catalogue offers a comprehensive record of the provincial distribution of Lepidoptera in Sweden, comprising information on moth occurrences in each province and the year that each species first colonised the province since 1974 [46]. These provincial data provided precise information about neighbouring species pools and yearly colonisation patterns in the regions. Records of moth colonisation dating back to 1974 have been collected for both provinces [46], with traits documented for all observed species [24]. The variable ‘number of years since provincial colonisation’ was calculated by subtracting the first year of observation for each province from 2020.

### 2.4. Data Analysis

All statistical assessments were executed using R software, version 4.3.0 [47], and the maps were created in ArcGIS Pro version 3.1 (ESRI, Redlands, CA, USA). To ascertain which provinces and trap sites captured the most diverse moth species, we scrutinised differences in the community composition between the sites using the Jaccard similarity index. This index was calculated as the number of species shared between the two sites (a) divided by the sum of (a), (b), and (c) where (b) and (c) were the numbers of unique species recorded at each site [48]. The results were illustrated through Venn diagrams using the ‘VennDiagram’ package (version 1.7.3) [49].

In order to provide a more comprehensive and interpretable representation of the biodiversity, we employed two modified diversity indices, both of which are classified as Hill numbers. Specifically, we used the inverse Simpson diversity index [50] and the exponential Shannon diversity index [51] These indices were applied to data from both trap locations, allowing us to perform a comparative analysis of the species diversity between the two sites. These indices were calculated using the ‘vegan’ package (version 2.6.4) [52] and accounted for both species’ richness and evenness [53]. A higher diversity index corresponds to a more diverse community.

We employed an individual-based abundance rarefaction methodology using the ‘iNEXT’ package (version 3.0.0) [54] to assess how well our sampling captured the species at each site. This analysis enabled us to visualise the species accumulation as a function of the number of individuals sampled and thus infer the efficiency of the traps and the saturation of our sampling [55].

The composition of moth communities between different sites and sampling periods was assessed using a non-metric multi-dimensional scaling (NMDS) analysis. For this purpose, we created a matrix of the species abundance-by-sampling period where each sample represented a trap catch with the number of individuals caught per species per trap catch, thus portraying the species composition from March to November across 46 unique samples. The NMDS analysis used the ‘vegan’ package (version 2.6-4) [56], employing the Bray–Curtis dissimilarity measure and default settings (three dimensions, stress = 0). A stress value of less than 0.10 indicates a well-represented community within these dimensions. To mitigate the impact of hyper-abundant species, we applied a square root transformation to the abundance data. We used the Permutational Multivariate Analysis of Variance (PERMANOVA, function adonis2 in R package vegan) to ascertain significant relationships between the species composition and the two trapping sites [56]. 

To analyse the impact of the number of years since provincial colonisation on the occurrence and abundance of moth species in these regions, we constructed a generalised linear model (GLM) with the ‘number of years since provincial colonisation’ and ‘site’ as the main predictors. This model was fitted with a Poisson error distribution to appropriately model the count nature of our response variable (abundance). To ascertain the significance of our predictor variable, an analysis of variance (ANOVA) was conducted using the ‘Anova’ function in the ‘car’ R package (version 3.1.2) [57]. Only species recorded in the trap and recorded since 1974 in the province were included in the GLM. 

To ascertain if certain species traits correlate with moths’ presence in the traps situated in human-influenced landscapes, we developed a generalised linear mixed model (GLMM) with a binomial error distribution. The GLMM was implemented using the ‘glmmTMB’ package (version 1.1.7) [58]. The dependent variable in our analysis was the presence of each species in the trap, represented as a binary variable. For each province, each species was classified into one of two categories: ‘present’ (assigned a value of 1) if the species were captured in the trap or ‘absent’ (assigned a value of 0) if the species was not found in the trap but was recorded in the species pool of the respective province.

The species pool represents the species known to inhabit the province where the trap was located which, thus, theoretically, could have been captured in the trap. This definition of the dependent variable allowed us to investigate not just the characteristics of species that were captured in the traps but also how these characteristics might differ from the overall set of species that could have been captured (i.e., the species pool). The model incorporated five species traits as independent variables: ‘colour pattern variation’, ‘habitat preference’, ‘length of flight period’, ‘host plant specificity’, and ‘overwintering stage’. Simultaneously, the model treated ‘province’ as a random effect to account for the potential non-independence of observations within these clusters. Additionally, we performed an ANOVA as described above. This method allowed us to investigate the statistical significance of each species trait in explaining the moth presence in human-influenced landscapes while controlling for the effects of the other traits. We calculated the odds ratios by exponentiating the estimates of the fixed effects. An odds ratio represents the multiplicative change in the odds of a moth being present for each unit increase in the respective species trait, assuming all other variables are constant [59]. For instance, an odds ratio of greater than 1 suggests that the odds of a moth’s presence increase as the trait value increases. In contrast, an odds ratio of less than 1 suggests the opposite relationship. The results were visualised using the ‘ggplot2’ (version 3.4.2) [60] and ‘gplots’ (version 3.1.3) [61] R packages. 

## 3. Results

### 3.1. Species Richness, Abundance, Rarefaction Curves, and Diversity

In total, 776 species were recorded across the two provinces, with 684 known species from Västergötland in 2019 and a corresponding number of 756 for Småland. A total of 664 species were shared between the two provinces. However, there were notable differences in the number of unique species in each province; Småland housed 91 unique species, whereas Västergötland was home to 20 unique species (Figure 1). The Jaccard index, a statistical measure of similarity between the two provinces, was 0.86, indicating a high degree of species overlap between these provinces. 

We recorded 27,797 individuals across 371 moth species from the two trapping sites in Västergötland and Småland. A comparison of the species recorded in our traps with those listed in the taxon-specific national catalogue revealed that 44% and 28% of the total species occurring in the surrounding province were found in the traps at Borås and Kalmar, respectively. In Borås (Västergötland), we trapped 20,491 individuals representing 301 species, while in Kalmar (Småland), we trapped 7306 individuals across 209 species. The distribution of species richness across the taxonomic families demonstrated notable variation between the two locations. In Kalmar, the family Noctuidae accounted for a substantial 55% of all captured species, while Geometridae represented 29%. In Borås, Noctuidae represented 41% and Geometridae constituted a significant proportion at 36%. The three families exhibiting the highest provincial record representation were Noctuidae, Geometridae, and Erebidae. Specifically, the Noctuidae family stood out with a significant representation of 53 provincial records across 45 species. This was closely followed by the Geometridae family, which accounted for 34 provincial records from 29 species. The Erebidae family was comparatively less represented, contributing only nine provincial records from six species. A detailed breakdown of the number of individuals per species, family, province, and trap is provided in Appendix A.

A comparison of the species richness between the sites revealed that 139 species were shared, with 70 species being unique to Kalmar and 162 species being unique to Borås (Figure 1). Among the recorded species, ten were red-listed, with five species being represented in Borås and five in Kalmar (Appendix A). The Simpson and Shannon diversity indices showed higher species diversity in Kalmar (Simpson = 8.12 and Shannon = 24.7) compared to Borås (Simpson = 3.25 and Shannon = 15.5), likely due to the higher species evenness at the former site. The shapes of the rarefaction curves indicate that the sampling captured most of the species present at each site (Figure 1). In Borås, *Eilema depressa* (55%), *Orthosia gothica* (5%), and *Lymantria monacha* (3%) were the most abundant species. In Kalmar, *Luperina testacea* (31%), *Xestia xanthographa* (10%), and *Orthosia cruda* (7%) dominated. Most species were rare, with singletons constituting 12% of the total catch in Borås and 30% in Kalmar. In Borås, species represented by five or fewer individuals accounted for 37% of the total, while in Kalmar, it was 58% (Appendix A). Non-metric multi-dimensional scaling (NMDS) achieved a satisfactory representation of dissimilarity in reduced dimensions (stress value of 0.086) (Figure 2). The PERMANOVA analysis, concurrently, indicated that the trap site (specifically, the regions of Borås and Kalmar) significantly influences moth assemblages (*p*-value = 0.005, F = 2.52, R^2^ = 0.053). 

### 3.2. Provincial Colonisation Rates and Species Shared between Province Records and Light Traps

In total, 628 species were known to exist in Västergötland in 1974 and 684 in 2019, and the corresponding numbers for Småland were 685 and 756. The discovery rates of new species in Västergötland and Småland increased between 1974 and 2019, with average rates of 1.09 species/year and 1.44 species/year, respectively (Figure 3). The results of the GLM revealed significant effects of the year, province, and their interaction on the number of moth species discovered (Figure 3, Table 1). In total, 56 species have colonised Västergötland since 1974, and 71 have colonised Småland. In Västergötland, a small proportion of the species found in the trap (4%, representing 12 species) is considered to be recent arrivals to the province, having colonised the area post-1974. These species account for a minimal part, approximately 1% (262 individuals), of the total number of moths captured. In contrast, in Småland, 5% of the species (11 species) found in the trap are recent colonisers, contributing to a more significant proportion, about 7% (509 individuals), of the total moth catch. At the Borås site, the most abundant recently (>1974) colonised species was *Miltochrista miniata*, with an abundance of 196 individuals. This species colonised the province of Västergötland in 2010. Another species, *Hypomecis punctinalis*, while less abundant with 19 individuals, has been in the province for a shorter period, having colonised the area in 2014. For the Kalmar site in Småland, the most abundant recently colonising species was *Hoplodrina ambigua*, with a remarkable abundance of 337 individuals. This species colonised the province in 2009. The second-most recently colonised species in Kalmar was *Cryphia algae*, found in 118 individuals colonising Småland province in 2017.

The number of years since colonisation and the trapping site significantly impacted the number of individuals of each species present in the trap (Figure 4, Table 2). The GLM revealed that, for each additional year since colonisation, there was a significant decrease in the number of individuals present (Figure 4, Table 2). Furthermore, the number of individuals of recently colonising species was significantly higher at the Kalmar site than at the Borås site. There was also a significant interaction effect between the years since colonisation and the location, indicating that the decrease in the number of individuals over the years differed between the two locations. Specifically, the decrease over time was more pronounced at the Kalmar site (Figure 4, Table 2).

### 3.3. Comparisons of Species Characteristics in the Trapping Sites and the Surrounding Regional Species Pool

Our generalised linear mixed model revealed significant relationships between the probability of a species being captured in the trap and the five examined traits: colour pattern variation, habitat preference, length of flight period, host plant specificity, and overwintering stage (Figure 5, Table 3). The colour pattern variation had a significant positive relationship with the likelihood of species being trapped. For every unit increase in the colour pattern variation, the odds of a species being trapped increased by 1.33 (*p* = 0.001). This suggests that species with a greater colour pattern variation are more likely to be found in traps (Figure 5, Table 3). Regarding the habitat preference, generalist species had higher odds of being trapped than species with a specific habitat preference, with an odds ratio of 1.29 (*p* = 0.07). This means that generalist species are 1.29 times more likely to be found in the trap than species associated with forest habitats. Conversely, species preferring open habitats were less likely to be trapped, with an odds ratio of 0.50 (*p* ≤ 0.001), implying that these species are roughly half as likely to be trapped compared to species associated with forest habitats (Figure 5, Table 3). The length of the flight period also demonstrated a significant positive relationship with the probability of being trapped. For every unit increase in the length of the flight period, the odds of a species being trapped increased by a factor of 1.07 (*p* ≤ 0.001) (Figure 5, Table 3). The host plant specificity of a species significantly affected its likelihood of being trapped. For each unit increase in the host plant specificity, the odds of a species being trapped increased by a factor of 2.09 (*p* ≤ 0.001), implying that species with a lower degree of host plant specificity (i.e., host plant generalists) are more likely to be trapped (Figure 5, Table 3). The overwintering stage showed significant effects on the trapping probability as well. The odds ratios for the imago, larva, and pupa stages were 0.27 (*p* = 0.001), 0.54 (*p* ≤ 0.001), and 0.49 (*p* ≤ 0.001), respectively, indicating that species overwintering as imago, larva, or pupa are less likely to be trapped compared to those overwintering as eggs (Figure 5, Table 3). 

## 4. Discussion

Our study explored patterns of species richness, diversity, and community structure at different spatial scales, as well as colonisation rates, abundance patterns, and the predictive power of species traits. We showed that the colonisation rate was statistically significantly higher in Småland compared to Västergötland. Furthermore, the abundance of species that had colonised the provinces since 1974 decreased with an increasing number of years since provincial colonisation. Importantly, our finding that the presence of moth species in traps could be predicted based on specific species traits in relation to the neighbouring species pool in the provinces underscores the value of ecological niches, requirements, and tolerances in understanding species distribution patterns. Indeed, in the Anthropocene epoch, where human influence has become a dominant force impacting our environment, rapid changes in flora and fauna at various scales are of escalating concern [62,63]. From global climate change driving broad-scale biogeographic shifts [64] to localised anthropogenic disturbances reshaping community compositions [65], these human-induced transformations exert significant pressure on the biodiversity.

The significantly higher colonisation rate observed in Småland compared to Västergötland implies that Småland may serve as a more accessible gateway for a multitude of factors such as the habitat quality, availability of food resources, climatic conditions, and species interactions, which could influence the likelihood of being exposed to species extending their ranges from countries south of the Baltic Sea. Furthermore, being more southern, Småland will naturally receive a higher influx of species seeking warmer habitats. This scenario is becoming increasingly common in the face of ongoing climate change. However, it was intriguing that the difference in colonisation rates between the two provinces was not as pronounced as expected (1.4 vs. 1.1 species per year in Västergötland) (Figure 3). This implies that numerous species are effectively expanding over substantial distances into regions of Sweden that are not directly adjacent to the Baltic Sea. This trend is concurrent with reports across northern Europe of numerous species expanding their northern range limits, a clear response to rising global temperatures [1,2,17,22,66]. Our study did not indicate any clear sign of increasing colonisation rates over time. Instead, our data showed a fit line that adhered closely to the data points, hinting at a constant colonisation rate in both provinces since 1974. This might indicate that species are steadily moving northwards across Europe [67]. In the Kalmar trap, we documented the presence of a species novel to the province of Småland, *Conisania luteago*. Since then, this species has expanded across southern Sweden [68].

Our finding that the abundance of recently colonising species decreased with the number of years since colonisation presents a fascinating and unexpected trend. One explanation is that when species first colonise new areas, they can exploit the novel environments efficiently, thus enabling the build-up of large populations. This pattern might not hold for all colonising species. Still, at least some appear to be capable of establishing substantial populations before eventually reaching an equilibrium with fewer individuals [69]. The recently colonised species *Hoplodrina ambigua* reached 337 individuals in Kalmar and *Miltochrista miniata* had 196 individuals in Borås. This underscores new species’ potential to become established and abundant. Recently established species can occasionally develop into non-native invaders and significant pest species [70,71]. In this study, the recent arrivals to the provinces made up 1% of the total abundance of trapped specimens in Borås and 7% in Kalmar. These figures suggest that a considerable proportion of the moths, particularly in Kalmar, are new arrivals. This proportion is likely to continue to increase in the future [1,13]. Indeed, of the 93 species that have colonised the provinces since 1974, 11 were detected in the Kalmar trap and 12 in the Borås trap, as represented in Figure 4. Four species have colonized the provinces since 1974—*Anorthoa munda*, *Noctua janthe*, *Peribatodes rhomboidaria*, and *Watsonalla binaria*—and were also present in both trap locations. This suggests that only a select few of the expanding species have extended their ranges and become so abundant that they were detected in both our traps. Contrasting these newer colonisers with species with a long-established presence in the provinces revealed some intriguing differences. The most abundant species (overall) were *Eilema depressa*, with 11,174 individuals recorded in Borås and only one individual recorded in Kalmar, and *Luperina testacea,* with 2269 individuals in Kalmar and seven in Borås. Clearly, these two species can develop populations reaching thousands of individuals and have been present in the provinces for as long as records have existed [72]. This long-term presence indicates these species’ successful integration and adaptation to the local habitats, leading to their superabundant status at anthropogenic sites. It would be compelling to integrate extant data pertaining to moths from diverse traps and regions to ascertain whether our findings are general or not [22,66,73,74].

The average species assemblage significantly differed between the two sites across the year (Figure 2). Despite this influence, however, it is noteworthy that it only accounted for a modest 5.3% of the variation within the dataset, a figure that, while statistically significant, warrants further scrutiny to more comprehensively understand the underlying dynamics at play. This might suggest that while the species assemblages at both sites share many species in common (hence the overlap), the relative abundances of these species are different enough between the two sites to result in a statistically significant difference overall. Our results reveal that the abundance, species richness, and diversity exhibit contrasting patterns, which appears counterintuitive, considering that these metrics often demonstrate positive correlations [75,76]. Specifically, our data from Borås illustrated a community dominated by a few species, which reduced the diversity despite having a higher species richness compared to Kalmar, which exhibited higher Shannon and Simpson diversity indices. This discrepancy in diversity indices could be attributable to the dominance of a few forest-associated species in Borås and lower light pollution levels compared to Kalmar [77]. This observation underscores the multifaceted nature of biodiversity and highlights the importance of comprehensive assessments of species richness to gain a more nuanced understanding of biodiversity patterns. Moreover, our findings suggest that there is greater similarity in species communities at the provincial level compared to at the local level, implying that larger areas have more community similarity than local sites within these large areas. Intriguingly, despite the significant geographical distance and varying environmental conditions, the anthropogenic impacts on the two sites manifested in a surprisingly high level of similarity between the sites [78]. These findings accentuate the complex interplay between biodiversity and anthropogenic influences and underscore the critical role of the spatial scale in ecological investigations [79].

Our study highlights that moth communities in anthropogenic trap sites are not mere random subsets of the potential colonising species from neighbouring species pools within respective provinces. Instead, the differences in species traits suggest a prominent role in spatial sorting and species filtering, indicating that deterministic processes significantly contribute to the shaping of the community assembly, beyond the roles of any stochastic events [80]. Our findings specifically suggest that anthropogenic influences may initiate a substantial filtering process that favours generalist species, although it must be acknowledged that the increased abundance of generalists may simply stem from their ability to utilise a larger fraction of resources compared to specialists [81]. These species tend to exhibit high intraspecific variability in colour patterns, long adult flight periods, and overwintering as eggs, along with being habitat and host plant generalists. That species with more variable colour patterns were over-represented in the light traps at the anthropogenic sites is in agreement with our prediction and probably reflects the association of polymorphic and variable colour patterns with broad niches, an improved ability to cope with heterogeneous and changing environmental conditions [9,12,17,24,25,41,42], and a reduced predation risk [43]. Interestingly, our study also suggests that the capacity to overwinter as eggs plays a significant role in moth survival within anthropogenic landscapes. Moths overwintering as eggs may fare better in these environments due to a decreased susceptibility to predation and enhanced resilience to environmental stressors, including desiccation, temperature fluctuations, and pollutants [9,82,83]. This advantage is likely due to the eggs’ protective structure and limited direct interaction with the environment compared to other life stages. This trait-driven selection implies that as anthropogenic influences and biotic homogenisation progress, we can expect future communities to contain fewer species and include larger proportions of generalists. Our study reinforces the concept of ‘anthropogenic exploiters’—species with specific life history traits, such as generalist behaviours, a high reproductive output, or adaptability to novel environments, which are more likely to thrive in any landscape significantly altered by human activity. As such, some species decline in abundance and range while others increase, pointing towards a large biogeographical reshuffling of species communities [13]. This transformation is driven by multiple pressures, such as the land use intensification, artificial light pollution, soil eutrophication/nitrification, urbanisation, and increasing temperatures [84], affecting all species, underscoring the irreversible nature of biodiversity loss and posing a significant challenge for humanity. Hence, understanding the implications of these changes and implementing measures to mitigate them should be priorities.

Our study identified ten red-listed moth species, offering important ecological insights into the surveyed regions (Appendix A). Some species, such as *Craniophora ligustri* and *Hepialus humuli* in Borås, are associated with deciduous forests and grasslands. In contrast, species found in Kalmar, like *Amphipoea crinanensis* and *Dicycla oo*, are drawn to oak forests and garden plants. This red-listed species decline highlights the impact of the disappearance of traditional low-intensive agricultural practices, which once fostered rich biodiversity. They now face anthropogenic threats such as habitat loss and fragmentation. For instance, *A. crinanensis* struggles with the overgrowth of formerly grazed wetlands and meadows, while *C. ligustri* faces the threat of ash dieback. 

While our study provides insights into biodiversity dynamics and the role of species traits, it does have a few limitations. Firstly, relying on trap data to understand the species distribution may lead to biases, since certain species may be under-represented if they are less likely to be trapped. Additionally, our study only considered two provinces, so our findings might not be generalisable to other regions with different climatic or geographical conditions. Future studies should validate and expand our findings by incorporating larger spatial scales, taxonomic groups, and larger areas. Further research could also focus on understanding the individual contributions of different ecological processes that shape the species diversity and richness. This might involve the investigation of the habitat quality, interspecies interactions, and temporal patterns in species occurrence. Finally, integrating a functional trait perspective could provide a deeper understanding of how species traits are linked to their distribution and abundance in anthropogenically altered environments.

## 5. Conclusions

Our study has shed light on the complex dynamics of moth communities, showing the influences of species traits, local habitats, and larger-scale anthropogenic factors on the species diversity, abundance, and community composition. That only 44% and 31% of the present species pools occur at local sites, respectively, is worrying, given that moths contribute significantly to various ecosystem services [85,86,87]. The observed trends suggest a potential severe future biodiversity decline and biotic homogenisation. These changes could have profound implications for community dynamics and ecosystem functionality. Understanding the role of species traits, especially for those succeeding in human-modified landscapes, is critical for future biodiversity research and conservation efforts. The trait-based approach can provide deeper insights into predicting which species are more likely to succeed under ongoing environmental changes. The stark reality of our findings lies in the ongoing transformation of biodiversity, a process driven by pressures such as artificial light pollution, soil eutrophication, intensified agriculture, and urbanisation. This shift in environmental conditions affects all species and underlines the irreversible nature of biodiversity loss, making it one of humanity’s most daunting challenges today. Therefore, we must understand these transformations’ implications and urgently implement informed mitigation strategies.

## Figures and Tables

**Figure 1 insects-14-00702-f001:**
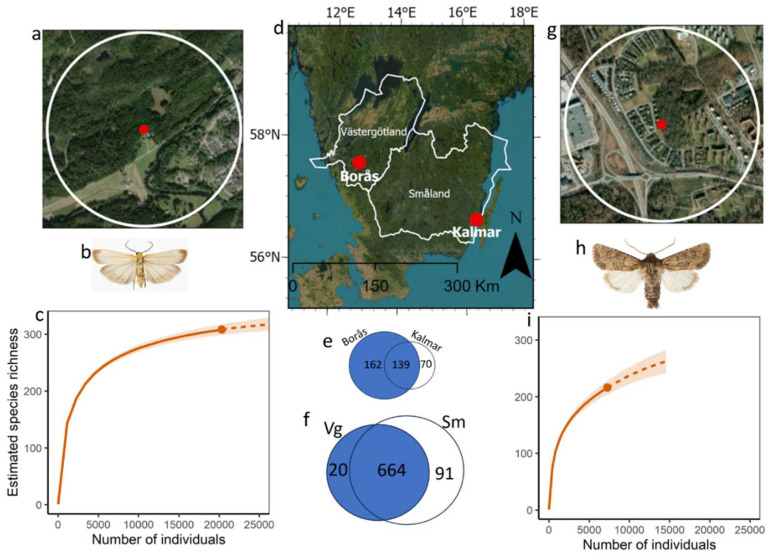
Overview and comparison of moth trapping sites as red dots and species richness in the provinces of Västergötland and Småland, Sweden (**a**). Satellite view of the moth trapping site in Borås, Västergötland, with the local area within a 1 km radius indicated by a white circle (**b**). Depiction of *Eilema depressa*, the most abundant moth species recorded at the Borås site (**c**). Individual rarefaction and extrapolation curves for the species richness at Borås, illustrating both the observed (solid lines) and projected (dashed lines) species richness. Shaded regions represent 95% confidence intervals (**d**). Geographical representation of the provinces of Västergötland and Småland in Sweden, with the locations of the respective moth trapping sites in Borås and Kalmar indicated by red dots. Provinces are outlined in white (**e**,**f**). Venn diagrams displaying the species overlap between the trapping sites (**e**) and between the provinces (**f**), with Småland abbreviated as Sm and Västergötland as Vg. Unique species counts are shown within the respective circles, and shared species counts are shown within the intersecting areas (**g**). Satellite view of the moth trapping site in Kalmar, Småland, with the local area within a 1 km radius indicated by a white circle (**h**). Depiction of *Luperina testacea*, the most abundant moth species recorded at the Kalmar site (photographs by Vladimir S. Kononenko) (**i**). Individual rarefaction and extrapolation curves for species richness at Kalmar, illustrating both the observed (solid lines) and projected (dashed lines) species richness. Shaded regions represent 95% confidence intervals.

**Figure 2 insects-14-00702-f002:**
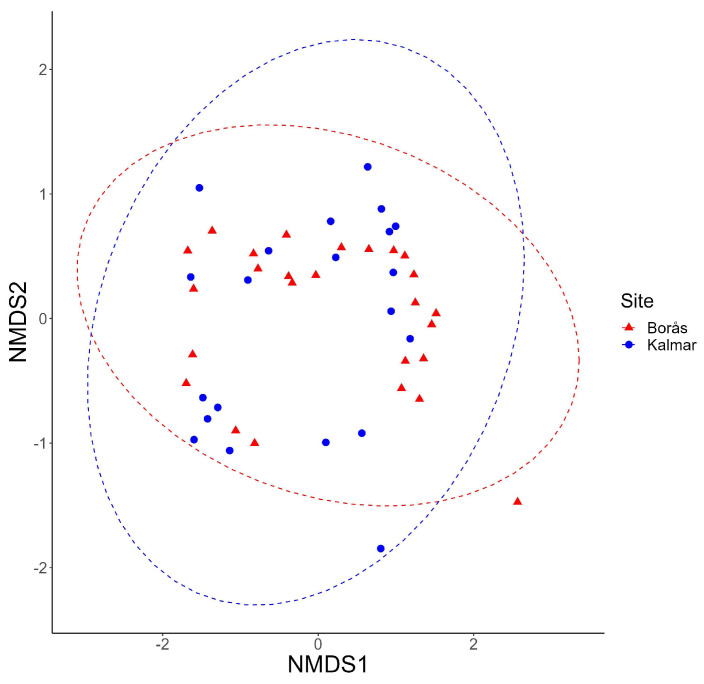
Non-metric multi-dimensional scaling (NMDS) analysis of species compositions across the two trapping sites in Borås and Kalmar. Each point on the plot corresponds to a discrete sampling event from a specific site, as designated by its respective colour. The generation of this plot was based on a Bray–Curtis dissimilarity matrix utilising root-squared species abundance data. Blue dots represent the Kalmar site, and red triangles represent Borås.

**Figure 3 insects-14-00702-f003:**
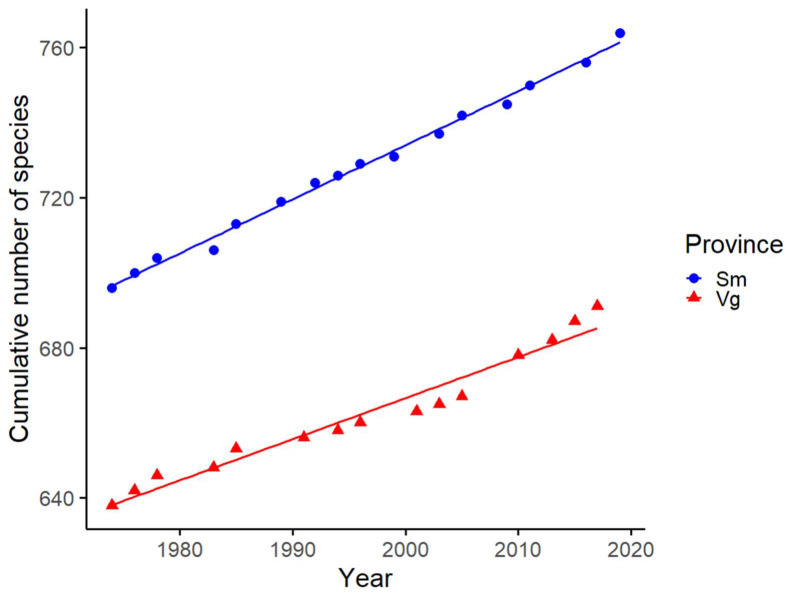
The cumulative number of moth species discovered over time. The scatter plot illustrates the changes in moth species discovered from 1974 to 2019 in Västergötland (Vg) and Småland (Sm) provinces. The coloured points represent the cumulative number of species discovered in each year, with blue points for Vg and red points for Sm. The solid lines indicate the linear regression model with the best fit for each province, showing the trend for the range expanding species over time. Linear regression equations: number of species (Sm) = −2151 + 1.44 × year and number of species (Vg) = −1525.7 + 1.09 × year.

**Figure 4 insects-14-00702-f004:**
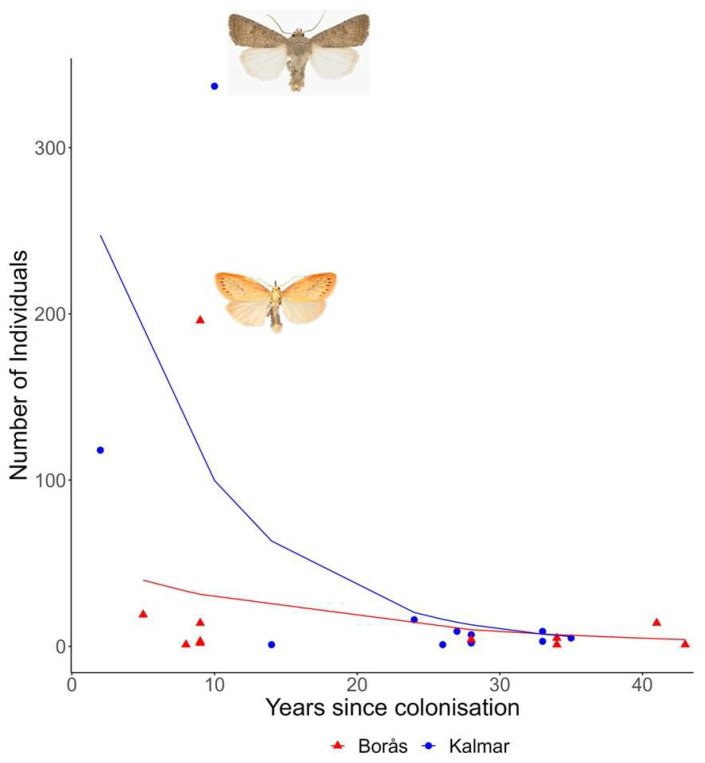
The relationship between the number of individuals from a single moth species captured in each trap and the number of years since the species colonised the province. The blue line and dots represent data collected from the Kalmar trap in Småland, while the red line and triangles denote information from the Borås trap in Västergötland. These lines depict the predicted correlation between the count of individuals and the number of years since colonisation, as indicated by a generalised linear model (GLM) utilising a Poisson distribution. The model accounts for differences in trends observed between the two sites. Regarding species that have colonised recently, *Miltochrista miniata* (colonised Vg in 2010, nine years ago, 196 individuals in the Borås trap) appeared in the highest numbers in the Borås trap, whereas *Hoplodrina ambigua* (colonised Sm in 2009, ten years ago, 339 individuals in the Kalmar trap) was most abundantly found in the Kalmar trap. These species are represented by images inserted directly into the graph, with photographs credited to Vladimir S. Kononenko.

**Figure 5 insects-14-00702-f005:**
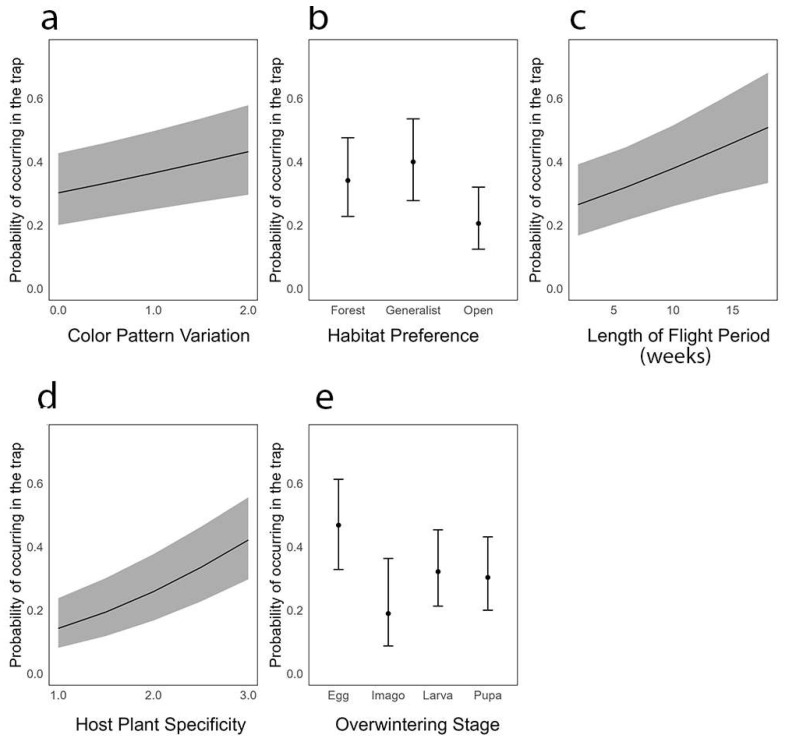
Visualisations of moth traits and their associations with the probability of occurrence in traps. Each panel (**a**–**e**) corresponds to one of five assessed traits. (**a**) Colour pattern variation (0 = no variation, 2 = highly variable), (**b**) habitat preference, (**c**) length of flight period, (**d**) host plant specificity (1 = specialist, 3 = generalist), and (**e**) overwintering stage. The line represents the fitted values from a logistic regression model for each trait, and the shaded areas represent the standard errors. For the categorical traits (**b**,**e**), each point is the fitted value at that category level, with error bars representing the standard errors.

**Table 1 insects-14-00702-t001:** Degrees of freedom (Df), likelihood ratio Chi-square (LR Chisq), and *p*-values from a generalised linear model (GLM) assessing the influences of the year and province on the number of species recorded in the province. The interaction term shows how the annual increases in species numbers are statistically significantly different between provinces.

Variable	Df	LR Chisq	*p*-Value
Year	1	1361.4	<0.001
Province	1	4740.8	<0.001
Year × Province	1	25.3	<0.001

**Table 2 insects-14-00702-t002:** Degrees of freedom (Df), likelihood ratio Chi-square (LR Chisq), and *p*-values from a generalised linear model (GLM) fitted to assess the influences of the number of years since colonisation and the location on the number of species. The GLM uses a Poisson distribution. The interaction term describes how the rate of change in the species count for each additional year since colonisation differs between sites.

Variable	Df	LR Chisq	*p*-Value
Years since colonisation	1	857.37	<0.001
Site	1	266.64	<0.001
Years since colonisation × site	1	34.37	<0.001

**Table 3 insects-14-00702-t003:** Degrees of freedom (Df), likelihood ratio Chi-square (LR Chisq), and *p*-values from the generalised linear mixed model (GLMM) using a binomial family to evaluate the effects of five different moth traits on the probability of occurrence in the trapping sites compared to the species pool in the province. The GLMM includes random effects for the province.

Variable	Df	LR Chisq	*p*-Value
Colour pattern variation	1	10.455	<0.001
Habitat preference	2	30.443	<0.001
Length of flight period	1	8.3084	<0.003
Host plant specificity	1	49.089	<0.001
Overwintering stage	3	22.741	<0.001

## Data Availability

The data supporting the findings of this study are available in Appendix A.

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
