# Peer review of "Anthropogenic Influence on Moth Populations: A Comparative Study in Southern Sweden"

_insects, 2023, doi:10.3390/insects14080702_

Round 1

Reviewer 1 Report

This paper combines two data sets that tackle range expansions of moths on southern Scandinavia. First, the increase of species lists in two provinces is analysed with respect to the influence of various species traits. Second, catches from two traps (one in each of these same provinces) are analysed from one single year. The logical connection of these two data sets is a bit difficult to grasp for the reader, since obviously rather different analytical methods had to be applied. Perhaps the authors could make it even more clear how these data sets integrate with each other?

Overall, I found the paper well written. The data clearly illustrate the massive and ongoing enrichment of the moth fauna of S Scandinavia in the course of climate change (and anthropogenic landscape transformation). The observed links with species traits are convincing, though similar findings have been made elsewhere (i.e. certain traits increase the likelihood of species to benefit from climate change or land-use change, e.g. J Mangels et al. (2017). Biodiversity and Conservation, 26, 3385-3405).

I have only a few rather technical suggestions for potential improvement during revision.

L 126: the light sources of the traps need to be specified and reported. Also, at what intervals trap catches were taken. Nightly? Weekly? At present it is impossible to understand in detail the nature of these trap samples.

L 169: exponential or logarithmic version of Shannon’s index? The exponential type is clearly to be preferred. Also, with the Simpson index. Please use the versions of both diversity metrics that are Hill numbers, as in your citation [47]!

L 208: did you apply a sqrt transformation to alleviate the influence of the few hyper-dominant species? This should probably be done before calculation of Bray-Curtis similarities.

L 209: should probably rather read: “Communities were considered well-represented in the dimensions if the stress value was <0.10”.

L 264: how could 4 and 5 red-listed species, respectively, sum up to 10?

L 265 and below: both diversity metrics not yet scaled as Hill numbers, please modify scaling!

L 290: in this NMDS ordination plot, no clumping according to trap sites is apparent. What precisely is each data point? The trap catch of one night, or week, or what? What does the first ordination axis refer to? Is that possibly a time axis (i.e. depicting parallel phenological shift in moth assemblages at the two trap sites, over the season)? This graph definitely needs better explanation. If there were no consistent differences between the two trap sites with regard to species composition, you could check that using a PERMANOVA test, with trap site as fixed factor. Given the strong overlap (see your Venn diagram), it might turn out that the smaller Kalmar sample is (apart from random noise in the data) just a subset of the larger Boras assemblage in the same year.

L 335: would not the relative contribution be more relevant on the y-axis, since the absolute size of the trap catch differed remarkably (almost three-fold) between the two trap sites? I presume numbers here refer to the sum over all moth species that were first seen in the respective province 1, 2, 3, ... n years ago. Please say that clearly. Was it really the case that at Kalmar only moth species present in the respective province since 2, 10 and 13 years were caught in your trap year, but none that occur in the province since, say, 9 or 11, or 12 or XXX years? Or, did I understand that graph wrongly?

L 373: what is the scale of the x-axis in the upper right panel? Weeks? In the lower left panel, perhaps the -axis title should rather read "range"? Here, the higher the x-axis value, the less specialized the species are, right?

Author Response

Reviewer 1

Comments and Suggestions for Authors

This paper combines two data sets that tackle range expansions of moths on southern Scandinavia. First, the increase of species lists in two provinces is analysed with respect to the influence of various species traits. Second, catches from two traps (one in each of these same provinces) are analysed from one single year. The logical connection of these two data sets is a bit difficult to grasp for the reader, since obviously rather different analytical methods had to be applied. Perhaps the authors could make it even more clear how these data sets integrate with each other?

** We understand the concern and have amended the text to clarify the rationale behind using two datasets and how they complement each other in our analysis. See the markedup copy for details.

Overall, I found the paper well written. The data clearly illustrate the massive and ongoing enrichment of the moth fauna of S Scandinavia in the course of climate change (and anthropogenic landscape transformation). The observed links with species traits are convincing, though similar findings have been made elsewhere (i.e. certain traits increase the likelihood of species to benefit from climate change or land-use change, e.g. J Mangels et al. (2017). Biodiversity and Conservation, 26, 3385-3405).

**We have added a more thorough discussion on how our findings compare and contrast with those from the mentioned literature and thank the reviewer for recommending a new reference we didn't know about.

I have only a few rather technical suggestions for potential improvement during revision.

L 126: the light sources of the traps need to be specified and reported. Also, at what intervals trap catches were taken. Nightly? Weekly? At present it is impossible to understand in detail the nature of these trap samples.

** We thank the reviewer for pointing out this lack of clarity. We have now specified the types of light sources used for the traps in the revised manuscript. We have also detailed the intervals at which trap catches were taken. We hope these additional details provide a clearer understanding of the nature of the trap samples.
L 169: exponential or logarithmic version of Shannon's index? The exponential type is clearly to be preferred. Also, with the Simpson index. Please use the versions of both diversity metrics that are Hill numbers, as in your citation [47]!

**We appreciate your suggestion and have revised our methodology accordingly. We used the exponential version of the Shannon's index and also the corresponding form of the Simpson index that are both Hill numbers. See the markedup copy for details.

L 208: did you apply a sqrt transformation to alleviate the influence of the few hyper-dominant species? This should probably be done before calculation of Bray-Curtis similarities.

**We followed your recommendation and applied a sqrt transformation to the data before calculating Bray-Curtis similarities. This has been updated in the Methods section and markedup copy for details.

L 209: should probably rather read: "Communities were considered well-represented in the dimensions if the stress value was <0.10".

**We thank the reviewer for spotting this and have changed it accordingly.

L 264: how could 4 and 5 red-listed species, respectively, sum up to 10?

**We apologise for the confusion. The correct number is 10.

L 265 and below: both diversity metrics not yet scaled as Hill numbers, please modify scaling!

**Following your advice, we have now scaled both diversity metrics as Hill numbers (see Lines 268-269).

L 290: in this NMDS ordination plot, no clumping according to trap sites is apparent. What precisely is each data point? The trap catch of one night, or week, or what? What does the first ordination axis refer to? Is that possibly a time axis (i.e. depicting parallel phenological shift in moth assemblages at the two trap sites, over the season)? This graph definitely needs better explanation. If there were no consistent differences between the two trap sites with regard to species composition, you could check that using a PERMANOVA test, with trap site as fixed factor. Given the strong overlap (see your Venn diagram), it might turn out that the smaller Kalmar sample is (apart from random noise in the data) just a subset of the larger Boras assemblage in the same year.

**We have added further explanation about the NMDS ordination plot in the revised manuscript. Each data point in the plot represents a trap catch. Additionally, we followed your advice and conducted a PERMANOVA test to check for differences between the two trap sites. See the markedup copy for details.

L 335: would not the relative contribution be more relevant on the y-axis, since the absolute size of the trap catch differed remarkably (almost three-fold) between the two trap sites? I presume numbers here refer to the sum over all moth species that were first seen in the respective province 1, 2, 3, ... n years ago. Please say that clearly. Was it really the case that at Kalmar only moth species present in the respective province since 2, 10 and 13 years were caught in your trap year, but none that occur in the province since, say, 9 or 11, or 12 or XXX years? Or, did I understand that graph wrongly?

**We appreciate your suggestion. We have chosen to implement a Generalised Linear Model (GLM) with a Poisson distribution and to discuss the relative values of recently colonised species in the discussion section. Considering that the relative contribution is modest, only constituting a few percentages, we deem it more illuminating to display actual numbers, which can reach several hundred individuals, yet still represent a marginal relative contribution.

L 373: what is the scale of the x-axis in the upper right panel? Weeks? In the lower left panel, perhaps the -axis title should rather read "range"? Here, the higher the x-axis value, the less specialised the species are, right?

** The scale of the x-axis in the upper right panel represents weeks. This has been clarified in the revised figure caption. In the lower left panel, you are correct; higher values on the x-axis indicate less specialised species. We have revised the figure legend and the figures accordingly. 

Reviewer 2 Report

This study utilizes two datasets – a historical dataset, reporting Macroheterocera data for the 45-year time frame on 1974-2019, and a recent dataset, obtained for 2020 from moth traps run in two locations. To me, these two datasets aim to answer different questions, yet they are intermixed in the aims/questions stated in the introduction, and in the results, which I found difficult to disentangle. I therefore encourage the authors to circumscribe the results from these datasets more separately (as far as possible). Also, it took me a good while in the beginning to figure out what the data basis of this study actually was - the simple summary and the abstract somewhat contradict each other in describing this.

I do not see how an in-depth discussion of the anthropogenic influences on species composition and diversity is possible (and useful) without having an additional trapping dataset from a much less human-impacted site to compare with. Generalist species might be as abundant in places less impacted then the Borås and Kalmar trap sites – but this has not been looked into. In the end that is the whole thing with generalists: they can utilise a larger fraction of resources than comparably more specialised species, and can thus be more abundant. The way the authors argue about the role of anthropogenic influences in the discussion therefore sounds a bit like a stretch to me, as I do not see it backed by clear data.

I would also be interested in seeing an analysis of phylogenetic patterns in the data: are certain families over- or underrepresented in species numbers in the traps? Are certain families more successful than others in establishing in the two provinces?

Why did the authors choose colour pattern variation as one of the five species traits investigated? There is no explanation for this choice, and the reasoning behind it, and I find it a very odd choice.

I also suggest to consistently use the same colour for the provinces in the Figures; I find the current state confusing (e.g. blue for Västergötland in Fig. 3, but red in Fig 4, and vice versa for Småland).

Please also see my comments in the PDF.

Author Response

Reviewer 2

This study utilises two datasets – a historical dataset, reporting Macroheterocera data for the 45-year time frame on 1974-2019, and a recent dataset, obtained for 2020 from moth traps run in two locations. To me, these two datasets aim to answer different questions, yet they are intermixed in the aims/questions stated in the introduction, and in the results, which I found difficult to disentangle. I therefore encourage the authors to circumscribe the results from these datasets more separately (as far as possible). Also, it took me a good while in the beginning to figure out what the data basis of this study actually was - the simple summary and the abstract somewhat contradict each other in describing this.

** We acknowledge the complexity of our two-dataset approach and understand your concern regarding the clear delineation of results. In our revision, we attempted to separate and clarify the results from the two datasets, ensuring they are linked to the relevant research questions (see marked up copy). We have also clarified the descriptions of the datasets in the abstract and summary to prevent any contradictions.

I do not see how an in-depth discussion of the anthropogenic influences on species composition and diversity is possible (and useful) without having an additional trapping dataset from a much less human-impacted site to compare with. Generalist species might be as abundant in places less impacted then the Borås and Kalmar trap sites – but this has not been looked into. In the end that is the whole thing with generalists: they can utilise a larger fraction of resources than comparably more specialised species, and can thus be more abundant. The way the authors argue about the role of anthropogenic influences in the discussion therefore sounds a bit like a stretch to me, as I do not see it backed by clear data.

**Anthropogenic Influences: Your point is well taken. In hindsight, we see how the absence of a less human-impacted control site may limit the depth of our discussion on anthropogenic influences. We have toned down our assertions regarding anthropogenic influences and clarified in the discussion that our results provide preliminary evidence of such influences, which need further investigation with additional data (see marked-up copy).

I would also be interested in seeing an analysis of phylogenetic patterns in the data: are certain families over- or underrepresented in species numbers in the traps? Are certain families more successful than others in establishing in the two provinces?

** This is a compelling suggestion. In our revised manuscript, we have incorporated family-level patterns, which offer insights into whether specific families are over- or underrepresented in the trap catches, and their success in colonising the provinces. Regrettably, there are only two to three species-rich families – Geometridae, Noctuidae, and Erebidae – while the remaining families are species-poor with few colonising species and low abundance. As a result, statistical analysis of different families proves unfeasible. We acknowledge the scientific value of such an analysis, and, given its significance, we aspire to address this in a separate, future study dedicated to a more detailed examination of phylogenetic patterns.

Why did the authors choose colour pattern variation as one of the five species traits investigated? There is no explanation for this choice, and the reasoning behind it, and I find it a very odd choice.

** We agree that we did not adequately justify investigating colour pattern variation as a trait. Our rationale was based on studies suggesting that moth colour patterns can influence their susceptibility to predation and thus their ability to establish in new areas. We have clarified this in the revised manuscript (see marked-up copy).

I also suggest to consistently use the same colour for the provinces in the Figures; I find the current state confusing (e.g. blue for Västergötland in Fig. 3, but red in Fig 4, and vice versa for Småland).

** Thank you for highlighting this oversight. We have consistently used the same colours for each province across all figures to avoid confusion (see all revised figures).

Please also see my comments in the PDF.

**Thank you again for your valuable feedback. We have carefully reviewed the comments you made on the PDF version of our manuscript. We've made corresponding revisions in our manuscript, marked with tracking changes in the revised copy. We sincerely appreciate the thoroughness of your review, which we believe has substantially improved our paper. If there are any further points or clarifications needed, please feel free to let us know.

We hope these revisions satisfactorily address your comments, and thank you for helping improve our manuscript.